# Comparison between Janus-Base Nanotubes and Carbon Nanotubes: A Review on Synthesis, Physicochemical Properties, and Applications

**DOI:** 10.3390/ijms23052640

**Published:** 2022-02-27

**Authors:** Sydney Griger, Ian Sands, Yupeng Chen

**Affiliations:** Department of Biomedical Engineering, University of Connecticut, Storrs, CT 06269, USA; sydney.griger@uconn.edu (S.G.); ian.sands@uconn.edu (I.S.)

**Keywords:** nanotubes, nanomaterials, CNTs, JBNts

## Abstract

Research interest in nanoscale biomaterials has continued to grow in the past few decades, driving the need to form families of nanomaterials grouped by similar physical or chemical properties. Nanotubes have occupied a unique space in this field, primarily due to their high versatility in a wide range of biomedical applications. Although similar in morphology, members of this nanomaterial family widely differ in synthesis methods, mechanical and physiochemical properties, and therapeutic applications. As this field continues to develop, it is important to provide insight into novel biomaterial developments and their overall impact on current technology and therapeutics. In this review, we aim to characterize and compare two members of the nanotube family: carbon nanotubes (CNTs) and janus-base nanotubes (JBNts). While CNTs have been extensively studied for decades, JBNts provide a fresh perspective on many therapeutic modalities bound by the limitations of carbon-based nanomaterials. Herein, we characterize the morphology, synthesis, and applications of CNTs and JBNts to provide a comprehensive comparison between these nanomaterial technologies.

## 1. Introduction

Among the current advances in engineering materials, nanotechnology has posed as a promising area of study for the purpose of enhanced drug and gene delivery, among other various applications. Nanoparticles come in a variety of morphologies and chemical compositions but are often spherical and have a diameter of ~50 nm or larger, limiting their ability for cargo delivery to cells or the extracellular matrix (ECM) [1,2]. As an alternative to spherical nanoparticles, nanotubes have generated a lot of interest over the past two decades, with carbon nanotubes (CNTs) showing some of the most potential due to their distinct and exceptional properties and wide array of applicability. There are two types of CNTs: single-wall carbon nanotubes (SWCNTs) and multiwall carbon nanotubes (MWCNTs). Their extreme strength and elasticity caused by the sp^2^ bonds between carbon atoms, electrical and thermal conduction, employable functionalization, and high tunability make for a novel material that has been used in composites, electronics, and, more recently, biomedical applications. CNTs were invented by Professor Sumio Iijima at Meijo University, and the graphene nanotechnology was awarded the 2010 Nobel prize in Physics [3]. However, while the nanotubes have small diameters that make drug or gene delivery into cells or ECM more achievable, CNTs also present significant limitations with their low solubility and difficult control over size and uniformity, contributing to a decline in biocompatibility and reproducibility. 

More recently, Janus-base nanotubes (JBNts) (also known as rosette nanotubes (RNTs), or helical rosette nanotubes), have demonstrated strong potential for tissue regeneration and overcoming of the issue of size and solubility, generating more possibilities for further advancements in biomedical applications. Janus particles are materials with non-centrosymmetric characteristics. The idea of using Janus particles for clinical use was an idea originally founded by De Gennes, a French scientist, who recognized the significance of an asymmetric reflection, coining the term “Janus” after the ancient Roman double-faced God [4]. The morphology and inherent magnetic and electronic properties are heavily dependent on the chemical composition and are being widely used in the biomedical field due to their low toxicity and biocompatibility with a wide variety of cell types [3]. JBNts, specifically, were inspired by DNA base pairs and are assembled using noncovalent bonding, making them biodegradable and biocompatible. They were inspired by the helical rosette nanotubes designed by Dr. Fenniri [5], modified by Drs. Yu and Chens to be more applicable to biomedical purposes [6]. In addition to cargo loading, the JBNt can be functionalized to enable targeted delivery [2], similar to CNTs. JBNts also have the ability to form a type of noncovalent linked nanoparticle called nanopieces (JBNps), which are multiple JBNt strands bonded together to form larger delivery vehicles, unlike CNT.

The purpose of this review is to compare the relatively new JBNt to the well-studied CNT in terms of their characterization, physicochemical properties, synthesis, and applications, bringing clarification to the distinct similarities and differences presented by both nanotubes.

## 2. Structure and Morphology of Nanotubes

### 2.1. Structure and Morphology of Carbon Nanotubes

To understand the structure of CNTs, it is important to analyze the structure of material from which CNTs are made: Graphene (Gr). Graphene is a two-dimensional, carbon crystal nanomaterial, or, simply, a crystalline two-dimensional graphite [3]. Having an isolated 2D material was only thought to be theoretically possible, as the crystal structure like graphene’s was believed to not be thermodynamically stable [7]. The carbons are bonded through sp^2^ hybridization, or sigma bonds, forming a hexagonal structure pattern throughout the graphene sheet on an atomic scale [8,9]. The applications of graphene since its discovery have been immense, with one application being the formation of CNTs. This graphene sheet forms CNTs when it is rolled into cylinders, creating a tubule with an extremely large length-to-diameter ratio (aspect ratio), and is capped on both ends by dome-shaped, half-fullerene molecules. SWCNTs only have one layer of graphene and a central hollow, while MWCNTs have multiple layers of graphene formed concentrically around the central hollow and have an interlayer spacing of 0.34 to 0.39 nm and an outer diameter of around 20 to 30 nm [10] (Figure 1). SWCNTs exhibit one-dimensional properties due to their nanoscale diameters, with the diameter of individual SWCNTs ranging from 0.4 to 2–3 nm, having only around 10 atoms comprising the circumference of the tube [9], and their length ranging in the micrometers [11].

There are three forms for a SWCNT to be assembled into—armchair, zigzag, and chiral—, and all three depend on the symmetry of the nanotube. The symmetry of the structure can be denoted by a set of indices (*n*,*m*), which are the graphene unit vectors. These vectors are the determining factor for the chiral vector, or the circumference of the tube, as well as the helical angle (χ), which is the angle between the axis of the tube and the edge of the graphene lattice [15]. These are important to determine the helicity of the CNTs and, thus, some of their physical properties. When *m* equals zero, χ equals 0°, which creates a zig-zag symmetry. When *n* equals *m*, χ equals 30°, forming an armchair symmetry. Finally, when the indices do not equal those of the previous two symmetries, they create chiral angles, forming chiral SWCNTs [15,16]. These symmetries significantly impact the physical properties of the nanotubes.

MWCNTs, on the other hand, only have two possible structural models. The first one is the Russian Doll model, where multiple sheets of graphene are formed into a nanotube and the middle tube has a smaller diameter than the outer tube. The second model is the Parchment model, which resembles a piece of rolled up parchment paper, as a single graphene sheet is wrapped around itself [17]. 

The ability for CNTs to be functionalized with different proteins or cargo loaded can lead to alteration of the physicochemical characteristics, which is an important barrier to note, as properties important to the function of the CNT could be disrupted [18]. Functionalization entails the addition of an amino acid side chain, or functional group. 

### 2.2. Structure and Morphology of Janus-Base Nanotubes

Janus-base nanotubes can also be referred to as rosette nanotubes (RNTs) due to their rose-shaped structure after assembly. RNTs can be characterized into two types: K1 (K = lysine) and twin base RNTs (TDLs) [18]. 

For both structures, self-assembly occurs between the Watson–Crick base pairs, or nucleobases, which are involved in cellular processes such as DNA replication, transcription, and translation [19]. They are characterized by having the ability to recognize their complementary base pair with their two hydrogen bonds and the major and minor grooves containing amino acids that bind to a corresponding amino acid on the base pair [20]. When the two nucleobases bind, they form a base–wedge–base triad motif [21]. The bonded base pairs also have the ability to interact with nucleotide bases and amino acid side chains via two hydrogen bonds [22]. 

In a K1 structure, the base pairs assemble into a six-member rosette using 18 hydrogen bonds in total, resembling a 2D nano-rosette. While side chains have the ability to be customized, the first K1 RNTs used lysine as a side chain, with the side chain extending outside of the rosette structure. In a TBL structure, two sets of base pairs covalently bond using 36 hydrogen bonds, with the sets of base pairs being parallel to each other, and a side chain is able to connect to both pairs, similar to K1. For both K1 and TBL, the single layer rosettes can stack to create a stable tubular formation with a diameter of around 3.5 nm to 3.8 nm, respectively [18] (Figure 2). 

What are referred to as JBNt are the RNTs with the K1 structure, with only one base pair making up the building blocks of the rosette structure, or supramolecule. However, the structure is no longer called “K1”, as the side chain does not necessarily need to be lysine to be considered JBNt. The base units of JBNt are still the DNA-inspired base pairs, with success in using both guanine and cytosine (G^C) [2] or thymine and adenine (A^T) [23]. 

Similar to CNTs, chirality of the structure is important to note as it is a determining factor of the three-dimensional organization of the supramolecule, and, therefore, the nanotube. The stacked rosettes self-organize to form helical nano-rods, with a predefined helicity being influenced by the asymmetry of the base units [24]. Chirality, or the dissymmetry of an object, is determined by the arrangement of the atoms. It is a very important concept in terms of pharmacology, drug production, and the prediction of the drug’s reaction in the body, with chirality possibly being the determining factor of either producing the desired effect or being toxic or inactive [25]. Unlike CNTs, there is a lack of research performed on the stereochemistry of JBNt; however, Hemrax et al. noted the inherent chirality of the twin G^C building blocks of RNT, as well as the effect of added chiral moieties and environmental conditions on the chirality of the nanotubes during synthesis [26]. Future studies could further investigate the effects of the nanotube chirality in terms of its interaction with their cargo, whether it is a drug or therapeutic gene, or cells within the body.

## 3. Synthesis of Nanotubes

### 3.1. Synthesis of Carbon Nanotubes

While multiple techniques have been developed for the development of CNTs, the three procedures that are currently commonly used are the carbon arc discharge technique [27], the laser ablation technique [28], and the chemical vapor deposition (CVD) technique [29] (Figure 3). There are some advantages and disadvantages for each method, with the ability to control the growth mechanism during synthesis being the universal challenge, as this achieves the desired physical and electrochemical properties [30]. Between the synthesis of MWCNTs and SWCNTs, the fabrication of SWCNTs poses the most difficult process to control, as it requires a catalyst and has poor product purity compared to MWCNTs. There is great promise with these techniques, but the search for a highly reliable and controllable synthesis technique for SWCNTs is still being continued. 

#### 3.1.1. Chemical Vapor Deposition

Compared to the other fabrication methods, CVD is considered to be the most effective for a large-scale control of the physical properties of CNTs [33]. The first successful fabrication of carbon nanofibers with the use of CVD occurred in 1975 when Oberlin et al. pyrolyzed a mixture of benzene and hydrogen at a temperature of 1100 °C [34]. Since then, the use of this method to create CNTs has been highly studied and analyzed, satisfying the requirement for a reliable process for the controlled growth of large quantities of CNTs.

The chemical vapor deposition method involves a two-step process with the assistance of a metal catalyst. The metal catalysts often used are nickel (Ni), iron (Fe), and cobalt (Co) [29]. The first step in the CVD method is depositing the metal catalyst onto a substrate, where the catalyst is then nucleated via chemical etching or thermal annealing, forming a site where particles can be deposited onto and allow them to grow [35]. The second step is the cleavage of a gas that contains a carbon atom, usually a hydrocarbon gas, that continuously flows through the catalyst nanoparticles. This decomposes the carbon source, successfully separating the carbon atoms from the gas in a tubular reactor [36] and dispersing it on the surface of the catalyst. The chemical vapor deposition reactors are designed to control the film thickness, crystal structure, surface morphology, and composition. It heats the substrate at a temperature of around 650–900 °C, which improves the flow dynamics and uniformity of the gas and dispersion [37].

For SWCNTs, specifically, the chirality, or helicity control, is determined by the hemispherical cap. During CVD for the growth of SWCNTs, the formation of the cap occurs at the nucleation of the catalyst [30], where the tube then follows a bottom-up approach by lifting off the substrate and elongating the tube. Therefore, controlling the cap of the nanotube is the key to controlling the properties of the CNTs developed. One of the disadvantages of CVD is that the metal catalysts required for synthesis are likely liquified from the high growth temperatures, resulting in a lack of control of the cap nucleation [38], and there are many parameters that can additionally influence cap nucleation [33]. This mechanism is not well understood, but a study by Rao et al. concluded that controlling the interfacial angles between the facets on the catalyst can potentially lead to helicity control during the growth of CNTs [31]. However, if the formation can be more easily controlled, the high yield of CNTs with little impurity from this method makes CVD a favorable fabrication technique.

#### 3.1.2. Carbon Arc Discharge Method

The oldest method for the fabrication of CNTs is the arc discharge method, with R. Bacon using it in the 1960s to create carbon fibers, or whiskers [39], which was later used to develop SWCNTs and MWCNTs fullerenes and eventually CNTs. The arc discharge method involves the use of two graphite electrodes, with an anode of either pure graphite or graphite with other metals. A direct current (DC) is established between the graphite electrodes under the exposure of an inert gas, such as helium or argon, and a pressure of around 500 torr [40]. A plasma is formed between the electrodes, producing small carbon fragments through the disruption of the graphite carbon networks of the anode, which are subsequently deposited onto the cathode [41]. After a reaction time of a few minutes, products are formed on various parts of the reactor: rubbery soot on the reactor walls, hard gray deposit at the end of the cathode, spongy soot surrounding the cathodic deposit, and web-like structures between the cathode and the chamber walls [39].

While MWCNTs do not require a metal catalyst, SWCNTs do, with the metal catalysts consisting of Fe, Co, and Ni, which is the same as CVD. When no metal catalyst is present, the spongy soot surrounding the cathodic deposit is not formed, and the resulting products are fullerenes, MWCNTs, and graphite carbon nanoparticles. In the presence of the metal catalyst, the products become MWCNTs and metal-filled MWCNTs, or FMWNTs, graphite carbon nanoparticles (GNP) and metal-filled GNPs, or FGNP, metal nanoparticles, and SWCNTs. Because of the mixture of products, purification is a necessary secondary step after the arc discharge production [39].

Using this method, the SWCNTs produced have a low yield number compared to CVD and a low level of purity, but have a high degree of structural perfection, with high crystallinity and no defects. This method has been determined to produce MWCNTs, but even with the elimination of a metal catalyst and the extra reaction byproducts, a high yield of pure MWCNTs still is a challenge due to the different reaction factors, such as pressure, current, and electrode composition, that not only affect the reaction products, but also can affect the morphology of the CNT [42].

#### 3.1.3. Laser Ablation Technique

The laser ablation process is commonly used to prepare nanoparticles [43], with SWCNTs being the primary carbon nanotubes produced using this method. While CVD and arc discharge methods have been successful in developing the nanotubes, there is still the challenge of fabricating pure SWCNTs [44]. Pulsed laser ablation was first used to develop SCWNTs by Guo et al. in 1995 [45] and Thess et al. [28] in 1996. This method of synthesizing SWCNT uses laser ablation involving a “double-pulse laser oven” process, where the laser varies between green and infrared, as well as the pulse widths within an Argon atmosphere [46]. The growth process takes place at around 1200 °C inside a double quartz tube reactor, where repeated laser pulses suspend carbon clusters in the inner tube and generate catalyst atoms and small carbon molecules, having the benefits of reducing amorphous or graphitic carbon production that could cause SWCNT impurity [47]. The nanotubes produced by Thess et al. demonstrated that the CNTs produced by laser ablation were in fact 90% pure, which is purer than those fabricated using the arc discharge method [28]. 

The mechanism for SWCNT synthesis is not well understood; however, Scott et al. proposed that the laser pulses heat the graphite-containing surface that also consists of the metal catalysts, which is extremely fine nickel and cobalt. This vaporizes the material, and the hot vapor expands and cools rapidly, and, as it cools, the carbon molecules and atoms condense to form larger clusters. The catalyst also condenses and attaches to the carbon clusters to prevent them from fully caging off, where this initial cluster continues to grow until and form the tubular structure of SWCNTs. The pulsed lasers excite the fullerenes generated by this method to emit C_2_, which feeds the SWCNT growth [45]. There have been studies suggesting that the structure of the SWCNTs can be controlled. For instance, when the diameter of the nanotubes becomes thinner when the laser pulse power increases, demonstrating that it is dependent on the laser power [27]. Some additional parameters include more laser properties, such as oscillation wavelength and repetition rate, the composition of the vaporized surface, and the pressure inside the chamber. 

Despite high potential for high yield of high purity [28], the main disadvantage with using the laser ablation method is that uniformity and straightness of the nanotubes synthesized is hard to control, and it is also not economically the best technique [46]. 

#### 3.1.4. Functionalization of CNTs

Functionalization of carbon nanotubes often is performed on CNTs, since the nanotubes do not have a chemical affinity to organic matrices, leading to difficulty dispersing CNTs into aqueous matrices [48] or creating a physical interaction between the nanotubes and the matrices due to the seamless surface of the CNTs [49]. Modifying the surface of CNTs makes it more desirable for a wider range of applications, specifically for biological applications, and the functionalization of them is almost as important as the synthesis of the nanotubes themselves [50]. Additionally, functionalization can help solve the already established problem of production methods synthesizing CNT with varying size and chirality and metallic impurities [51]. The two different approaches for functionalizing CNTs are chemical (covalent) and physical (noncovalent/van der Waals bonds) (Figure 4). For both method types, the process of functionalizing CNTs occurs after the synthesis of the nanotubes themselves [52]. 

Covalent functionalization utilizes the sp^2^ carbon armature to attach chemical moieties [54] such as metals [55], functional groups (OH, COOH, and NH_2_), and amino acids to the exterior surface of the CNT tubular structure [54]. This allows the physicochemical properties to be fine-tuned depending on the desired application. Bonds of sp^2^ hybridization are extremely stable, therefore the disruption of the bonds is necessary for functionalization, and one way to do so is by fluorinating CNTs, due to the high chemical reactivity [56]. The fluornation causes C-F bonds to form, making some sp^2^ bonds hybridize into sp^3^ bonds, thus weakening the bonds and providing substitution sites for functional groups to attach. Defect functionalization is also another covalent method, where oxidative damage creates defects in the nanotubes, which leave holes that can then be filled with oxygenated functional groups [57]. With this method, Heister et al. succeeded in creating triple functionalized SWCNTs, covalently attaching fluorescein and CEA antibodies to oxidized nanotubes [58]. However, covalent functionalization unfortunately causes inevitable defects, specifically the hybridization of sp^2^ to sp^3^, disrupting the integrity of the structure, which not only affects the mechanical properties, but also the transport properties [51]. 

Noncovalent functionalization solves the issue of the disadvantages of disrupting the conjugated system of the CNT structure, allowing for the retention of the intrinsic properties of the nanotubes. This method of functionalizing CNTs once again allows for the addition of functional groups to the exterior interface of CNTs by employing hydrophobic interactions through van der Waals forces, or π–π bonding [59]. The process of noncovalently functionalizing CNTs involves high affinity molecules, such as surfactants and aromatic molecules, which help to enhance the solubility of the CNTs for biological applications. Previous work by O’Connell et al. outlined the mechanisms behind a more efficient and effective SWCNT solubilization method involving the incorperation of water soluble polymers such as polyvinyl pyrrolidone (PVP) and polystyrene sufonate (PSS). By introducing water soluble polymers to the SWCNT solution, the hydrophobic interactions between SWCNTs and their polar solvent is reduced, therefore minimizing thermodynamic penalty while also eliminating the problematic aggregation of nanotubes in a traditional aqueous solution [60]. This is a major advantage in respect to the functionalization potential and utilization of the pristine SWCNT framework, since their surface is more readily accessible to modification. Woods et al. used aromatic rings, or benzene derivatives, to functionalize SWCNTs, and found that the nanotubes were able to absorb the functional group through the interaction of the π orbitals on the CNT and the benzene ring [61]. This π–π interaction can also be used to functionalize CNTs with polymers, such as cellulose derivatives and polypyrroles [62], which have been utilized for several applications [63]. 

There have been some difficulty fabricating nanotube-based molecular assemblies because the incorporation of highly engineered molecules on the nanotube surfaces has been problematic. This is usually caused because the molecules being functionalized onto the surface of the CNTs are incompatible or require a large excesses of reagent. A research group including Dr. Tomás Torres aimed to overcome this issue by using “click chemistry” and attaching zinc-phthalocyanine (ZnPc) onto the surface of SWCNTs using Huisgen 1,3-dipolar cycloaddition. Phthalocyanines (Pcs) are planar electron-rich aromatic macrocyles that act as excellent onor/antenna building blocks, and, after the nanotubes were functionalized with 4-(2-trimethylsilyl)ethynylaniline onto the SWCNTs and the subsequent addition ZnPc after, was shown to improve fabrication of more complex functional structures [64]. 

### 3.2. Synthesis of Janus-Base Nanotubes

While CNTs have been extensively studied for the past few decades, JBNt are relatively new, with the synthesis process for the structure coined “JBNt” only being patented in 2017 [6]. Because of this, the diversity in preparation and synthesis methods are fewer than the wide range of methods for CNTs. 

Zhou et al. synthesized the JBNt monomer according to the previously published patent [2]. The synthesis of the monomer alone involves the intermediate steps A1–A9, with the final step using A9 to yield crude JBNt monomers, where it is subsequently purified by high-performance liquid chromatography (HPLC). This results in a janus-base monomer consisting of DNA-base pairs adenine and thymine with a lysine side chain. As previously established in the section discussing the structure of JBNt, the prepared JBNt monomers self-assemble into six-member rosettes through hydrogen bonding to form a plane [63], which subsequently stack onto each other via strong π-stacking interactions [5], forming the nanotube structure (Figure 5). 

JBNt can also be synthesized to form larger drug delivery vehicles, such as Janus-base nanopieces (JBNps). Both Sands et al. and Lee et al. described the synthesis of JBNps that stems from JBNt, where the cargo is involved in the fabrication process [2,65]. The JBNt and the nucleic acid cargo self-assemble, with the JBNt surrounding and locking in the strands of cargo, which is siRNA in this study, to develop a multi-strand delivery system. Using sonication, the size of the nanopieces can be varied by separating the JBNps through agitation [2].

#### Functionalization of JBNt

Unlike CNTs, the functionalization occurs during the synthesis process of JBNt, with a dependency on the components and solutions used during the intermediate steps of the JBNt monomer synthesis. For the patented JBNt, the functional group is lysine. For JBNps specifically, there is great potential for additional surface modification, such as attaching targeting ligands onto the surface of JBNps to invoke a desired bioresponse or to target specific cells that are to be further studied for additional applications of JBNt and JBNps. Additional modifications can be made, such as Polyethylene glycol (PEG) coating to the exterior of JBNp vectors, to increase protection from degradation factors involved in systemic drug delivery. Further research on the functionalization of different molecules or side chains in order to test application efficiency and biocompatibility is needed in order to compare the range of functionalization of JBNts/JBNps to that of CNTs. Since JBNts possess a more complex structure than CNTs, modifications can be made to the fundamental chemical structure of JBNts to adjust physiochemical and electrochemical properties. Evidence of the functional evolution of janus-based nanomaterials has been characterized previously, demonstrating growth that this field has yet to realize [66].

## 4. Biomedical Applications

### 4.1. Drug Delivery

The pharmaceutical industry faces challenges regarding the intrinsic properties of potential drugs that could hinder the efficacy of their therapeutic effect. This is forcing them to develop drug delivery systems, with successful designs involving the use of nanotechnology due to their unique chemical and structural characteristics that allow for the targeted drug delivery, increased solubility, and increased stability of the drug [67,68,69,70]. 

#### 4.1.1. Drug Delivery via CNTs

One of the nanotechnologies studied as a drug delivery system is CNTs, helping to overcome the limitations posed by free drug administration and can even allow for combination therapy through the co-delivery of drugs [71]. CNTs have shown incredible success and potential for drug delivery [72], gene delivery [73,74], and vaccine delivery [75]. Using the functionalizing techniques, pharmaceutical entities can be covalently and noncovalently bound, or functionalized, to the CNT structure [67], such as anticancer, antiviral, or antibacterial agents that can be covalently functionalized to the nanotubes [76]. There has also been success in functionalizing CNTs so that they target specific cell receptors to be uptaken by those cells or to release their therapeutic agent at a targeted site [77]. Al-Jamal et al. modified MWCNTs with siRNA that targeted caspase-3 to treat brain tissue loss following a stroke, successfully demonstrating that the rats that had the functionalized nanotubes administered into their cortex had reduced apoptotic cells and neurodegeneration [78]. 

Unfortunately, while the potential and current uses for drug delivery remain optimistic, CNTs present as cytotoxic when absorbed into the cells’ intracellular space during the drug delivery process and can be heightened by poor dispersal or aggregation of the CNTs. Functionalizing the CNT can help decrease cytotoxicity during the cellular uptake process [79], but there is still a level of toxicity, especially when the concentration of CNTs increases. Characterizing the cytotoxicity profile of the CNTs, however, is a little more complicated and has not been well established, as there are many different factors that contribute to the toxicity of the nanotubes on the cells. CNTs can produce active forms of oxygen as a byproduct, as well as reactive oxygen species (ROS), which can damage cells. While cells have the ability to protect themselves against these oxygen forms in low concentrations, high levels of CNTs produce higher levels of the oxygen species, thus damaging the cells and causing cellular death. Literature and studies have inconsistencies regarding the toxicity of CNTs, with a dependency on concentration, synthesis, functionalization, electrochemical dopants, oxidative stress, etc. [80]. Dr. Emilio Pérez and Dr. Sofia Mena-Hernando have also discussed mechanically interlocked SWCNTs, which could be applied for drug delivery purposes. Mechanically interlocking redox-active anthraquinone onto SWCNTs provides a noncovalent and electrochemically stable nanostructure and stable oxygen reduction reaction in an aqueous solution, therefore potentially reducing the levels of ROS produced by the CNT during drug delivery [81]. Dr. Nazario Martín and Dr. Dirk M. Guldi’s book speaks on the extensive potential toxicity of CNTs, and it mentions that there is even the possibility for MWCNTs to induce clastogenic and aneugenic events, breaking DNA strands or eliminating or adding whole chromosomes. The authors also explain that toxicity is highly dose-dependent and toxic activity is highly dependent, which is ultimately caused by the defective sites in the carbon framework but is still very difficult to stigmatize defininite assumptions on this subject [82]. Therefore, for use of CNTs as a drug delivery vehicle, it is important to note that the cytotoxicity is varied between each study and cannot be deemed as a nontoxic delivery method for cellular uptake without further research on the specific study at hand. 

As previously discussed, the nanotubes have a hollow channel in the center, which can act as a “nanoreservoir” for drug delivery, allowing for the nanotube to perforate cellular membranes without cellular interaction with the drug itself until released at the targeted site. While conventional drug loading often occurred on the outer surface of the nanotube through covalent or noncovalent bonding, Luo et al. utilized the inner cavity for further advancements in drug delivery methods. When the caps of the CNTs are opened, the cavity can be filled with various components, such as proteins, peptides, small molecules, water, and nanoparticles, and more recently there has been research done on filling the CNTs with bioactive drugs. The convenient method of filling the reservoir requires the drugs or molecules to be suspended in a solution, which has posed a challenge due to the surface tension of the liquid reducing the loading efficiency. Another challenge is to deliver the drug to a targeted site without any leaking of the solution through the caps. Luo et al. used MWCNTs and sonicated them to shorten the length and open the caps and treated the nanotubes to be more hydrophilic in order to prevent the surface tension from decreasing the filling efficiency. They were filled with Dexamethasone (Dex) and were coated with PPy so that the nanotubes could release the drug upon an electrical stimulation [83]. Another study published by Dehaghani et al. analyzed the ability for end-capped SWCNT to internally host the anticancer drug, Isatin, and demonstrated that the nanotubes successfully trapped the drug cargo stabally inside the cavity [84]. The structure and intrinsic chemical properties of CNTs have led to a lot of medical and pharmaceutical advancements, with current ongoing study on this topic [85]. 

#### 4.1.2. Drug Delivery via JBNts

Similar to CNTs, the versatility of JBNt extends to an array of drug delivery applications, with their ability to encapsulate therapeutic agents and to be functionalized to achieve delivery at targeted sites. Song et al. utilized the tubular structure of the single-base and twin-base RNTs to load tamoxifen (TAM), a hydrophobic anticancer drug [18]. Water-insoluble, or hydrophobic, drugs are typically considered difficult to administer due to their inability to dissolve in an aqueous solution, which, in turn, decreases their drug availability [86]. The exterior surface of an RNT is hydrophilic and the interior canal is hydrophobic, allowing the hydrophobic drug to be hosted and shielded within the tube to allow for prolonged release, which was successfully exhibited during this study for TAM administration [18]. Hollow canal loading has also been achieved with hydrophobic drugs such as dexamethasone (DEX) to provide slow release of anti-inflammatory therapeutics [87]. The ability for RNTs, along with JBNts, to be functionalized with peptides [88,89] suggests that they have the ability to co-deliver factors, such as drugs and growth factors, simultaneously. Additionally, RNTs have been established to be biocompatible and bioactive, with properties that enhance protein absorption and cell adhesion [65,88], entailing a nontoxic drug delivery system that concurrently delivers therapeutic agents while promoting healthy cell regeneration and recruitment. 

The formation of Janus-base nanopieces through further synthesis of JBNt allows for a new drug delivery vehicle. Therapeutic gene delivery has been successful with the use of JBNt. RNA interference (RNAi) therapies, including siRNA, require a nanocarrier to shield the sensitive and negatively charged interfaces of the siRNA strands that make delivery into cellular membranes and ECM difficult [90]. A previously discussed approach for siRNA delivery consisted of the small interfering RNA being “sandwiched” between JBNt strands, with the short and slim morphologies of the delivery vehicles resulting in increased penetration to extracellular matrices [91,92]. This structure was eventually termed a “Nanopiece,” which has the diameter of a tubular structure, being ~20 nm, and a positive ionic charge that makes it uniquely qualified for penetrating the ECM for the purpose of gene delivery into hard-to-reach areas. It is currently being studied as a delivery method into cartilage ECM, which has been an extremely difficult task to perform due to the dense matrix structure [2]. Another study by Lee et al. used the JBNps to enter the cellular membrane via macropinocytosis to successfully deliver siRNA, where the JBNps can then escape from endosomes using a “proton sponge” effect (Figure 6). This has shown promise in replacing the use of cationic polymers, which usually have low biodegradability and biocompatibility, due to the JBNps DNA-mimicking chemistry and noncovalent bonds [92]. JBNps also express superior uptake efficiency and penetration similar to lipid nanoparticles (LNPs), such as lipofectamine. However, JBNps have the advantage of escaping the endosome, while LNPs struggle to do so. While LNPs have a lot of advantages in RNAi delivery, the NPs highlight the advantages of the LNP carrier, while also avoiding the drawbacks that LNPs and polymers present [92,93]. The functionalizability, exterior hydrophilicity, and formation of NPs hold strong promise in pharmaceutical applications of drug and gene delivery that is being further studied. Furthermore, current work suggests that, like CNTs, JBNts are not limited to the functionality of a single loaded cargo. Instead, JBNTs may be co-loaded with drugs, nucleic acids, and other entities that provide a multifaceted approach to boost therapeutic efficacy. Ongoing work has also shown that JBNts are not limited to small base pair nucleic acid cargos such as miRNA or siRNA. Tunable JBNt morphology and side-chain properties make them excellent candidates for larger JBNps loaded with mRNA. While many delivery vectors including CNTs and JBNts have demonstrated excellent potential for therapeutic RNA delivery, most materials are limited to smaller RNAs, which in turn limits their impact on gene expression regulation. The incredibly versatile self-assembly and layered encapsulation of JBNts can allow for the total protection and delivery of larger base pair RNAs into cells for transcription. This opens the door for additional therapeutic applications by means of positive transcription regulation [94]. Additional work must be done to explore JBNts’ potential for noncovalently bound conjugates for stimulated release. JBNts traditionally rely on charge interactions to encapsulate therapeutic elements, however their chemical structure allows for substantial electron dislocation and stimulation potential. This indicates a potential avenue for JBNts as a delivery vector capable of stimulated cargo release with no cytotoxic effect due to a completely noncovalently bound complex.

### 4.2. Electrical Conductivity

The ability for these nanomaterials to carry and distribute electrical charge adds additional therapeutic benefits in biomedical research. The high charge capacity of these nanomaterials allows for the synthesis of standalone or composite constructs capable of transporting electrons to and from target locations as a means of stimulation and/or recording. CNTs and JBNts both share similar methods of electron dislocation to allow for efficient standalone charge transfer and enhanced versatility.

#### 4.2.1. Electrical Conductivity of CNTs

The CNT’s high electrical conductivity is derived from the extremely low-defect chemical structure of graphene. The lattice network of sp^2^ hybridized carbon atoms that comprise a carbon nanotube are bound by a combination of sigma bonds between carbon atoms and π bonds above and below the graphene sheet. The overlapping distance between adjacent π orbitals creates a parallel path for electron dislocation and charge conduction. Due to the pristine structure of the crystal lattice, most electrical conductivity interference must come from extrinsic factors. Many of these factors include interaction between CNT and its substrate as well as interfacial phonons [95]. Although good in theory, realistic applications of CNTs take chemical dopants into consideration to modulate electrical and structural properties. These dopants must be considered prior to biomedical application due to the potential local cytotoxicity during acute and chronic timepoints.

CNTs can play a versatile role in addition to conjugation with growth factors and and small drug molecules. Their robust electrical conductivity makes CNTs viable candidates for neural interfacing applications such as stimulation and recording. The increased capacitance of CNT-coated electrodes results in decreased impedance and the capability for a greater voltage polarization window for electrical stimulation [96]. CNTs have a proven biocompatibility with neural cells in vitro and have demonstrated increased neural activity upon cell attachment. Greater focus has been directed towards the preparation and controlled properties of CNT films that interface with neural tissues, as toxic dopants may present longevity performance issues upon chronic simulation at the electrode–electrolyte interface [97]. CNTs have also effectively increased neural differentiation and maturation in combination with a softer composite-mediated ECM [98]. Due to their ability to mediate and encourage cell signalling and electrical activity, CNT-hydrogels have been shown to increase neural stem cell differentiation and maturation in vitro while simultaneously increasing synaptic conectivity and neural network activity. Electrically stimulated composites have also allowed for the controlled release of bioactive molecules such as nerve growth factor (*NGF*) to increase neural differentiation and maturation [99]. CNTs are excellent candidates to enhance interactions with previously described methods for electrically driven neural stem cell differentiation and maturation. In addition to their natural conductivity, substrate nanotopography provides a more complex physical network for cell connectivity and proliferation [100]. Enhanced neurite growth and differentiation have been seen using CNT-based scaffolds with and without electrical stimulation. These results have been hypothesized to be primarily driven by the activation of differentiation-associated signaling pathways that direct progenitor cell fate and function [101].

CNTs have also been shown to assist in cardiac tissue regeneration due to their ability to fascilitate electrical stimulation to hMSCs. In vitro demonstrations of effective hMSC differentiation towards a cardiomyocyte lineage have been performed by providing cardiomimetic electrical cues over the course of 2 weeks [102]. CNTs also demonstrate a favorable interaction with cardiomyocyte membrates, allowing for improved electrophysiological recordings of cardiac tisues obtained with CNT-based devices [103]. Improved electrical stimulation and recording helps to ensure protective measures against pathological hypertrophy of cardiac tissue; however, again, additional attention must be directed towards the synthesis and manipulation of CNTs to ensure minimal local cytotoxicity.

#### 4.2.2. Electrical Conductivity of JBNts

While CNTs have been extensively characterized in respect to their electrochemical properties, JBNts have drawn much less focus, and their potential for many biomedical applications remains unrealized. While CNTs are designed from pristine graphene, JBNts are comprised of aggregated G^C monomers, each containing a number of sp^2^ hybridized aromatic rings. Although CNTs and JBNts are similar in shape, JBNts aromatic ring systems lie above and below each other, extending parallel to the length of the nanotube. The space between these aromatic rings is occupied by π orbitals intersecting with the neighboring rosette, allowing electron dislocation to occur through and along the axis of the nanotube.

Some research groups have demonstrated the versatility within JBNts chemical structure by synthesizing tricyclic variants termed “J-Type” rosette nanotubes. These nanotubes expanded upon the existing G^C byciclic motif by adding an internally fused pyridine ring to form the tricyclic variant [104]. This J-Type variant promoted stronger and larger π–π interactions, established optimal interchromophoric distances, and increased electron delocalization.

JBNts maintain superior biocompatibility due to their DNA-mimicking chemistry and lack of covalent bonding. Coupled with their tunable electrical conductivity, their potential use in standalone and composite constructs remains unexplored, particularly in applications that require electrical stimulation and recording. By mitigating the cytotoxic risks associated with other conductive materials, JBNts provide a promising bio-friendly platform for interfacing with biological environments.

### 4.3. Scaffolding and Coating

In the field of tissue engineering, scaffolds and implant coatings are extensively studied and utilized to facilitate structural support and cellular adhesion and proliferation [90,105,106]. Nanoparticles are often incorporated into scaffolds and coatings to form composites with the purpose of increasing mechanical strength and surface properties to achieve the desired outcome out of the biomaterial [107]. Both CNTs and JBNts are currently being utilized as scaffolds and coatings for improvements in implanted scaffolds and biomaterial success [108,109]. 

#### 4.3.1. Scaffolding and Coating with CNT

The intrinsic mechanical and chemical properties of CNT make the nanotubes an ideal material with which to prepare scaffolds and coatings for biological implants. SWCNTs have an extremely high strength of ~37 GPa due to the sp^2^ bonds [16], making them stronger than steel and Kevlar [11], despite being so lightweight. Additionally, CNTs have a better thermal conductivity than diamond, making their applications extend much further than only biomedical purposes [110].

CNTs can be fabricated to form aligned and conductive scaffolds, which form a microenvironment that supports the growth and regeneration of cardiac muscle. Traditionally used scaffolds for engineered myocardial tissue consisted of polymeric materials with insulative properties that hindered the transduction of the cardiomyocytes’ electric signals. A CNT scaffold overcomes this obstacle due to the nanotube’s conductive properties that propagate the signal transduction, demonstrating greater cardiomyocyte recruitment and differentiation, with the added bonus of improving the structural integrity of the scaffold [111]. Furthermore, CNTs are utilized to coat scaffolds. Composites of different biomaterials and CNTs are typically what compose the coating and are used to prevent degradation and corrosion and adjust surface wettability [97]. The electrically conductive properties of CNT are also beneficial when being used as a coating. A PPy-CNT coating was used on a platinum/tungsten microelectrode for intracortical use in rats. This coating was shown to significantly reduce impedance and improve (SNR), demonstrating increased electrochemical performance of the electrodes [112]. While toxicity still poses a challenge for the use of CNTs, for interfacing purposes, there has been very little cellular death and actually have been shown to increase cellular adhesion and proliferation, indicating them as biocompatible [76]. CNT-based nanofiber coatings show increased osteoblast adhesion [113] as well as other cells due to the absorption of proteins onto the nanotubes, laying down a matrix for cell adhesion [114]. This biocompatibility of CNTs also makes them strong candidates to use for the nanoscale surface modification of implants to increase integration of cells and native tissue connection, and they have been considered for use on implants to treat Cervical Degenerative Disc Disease (CDDD), with the nanoscale modification aiding in improved clinical outcomes [115]. A main disadvantage of using CNTs for scaffolds and coatings is the lack of biodegradability of the nanotubes, but even when dissolving CNTs in a biodegradable matrix, such as chitosan, as shown, improved mechanical properties and cell activity [11]. Additionally, a previously described work by O’Connell et al. may help to address concerns realized by many CNS interfacing applications involving CNTs. Electrically conductive scaffolding for interfacing applications such as intracortical microelectrodes oftentimes utilize SWCNT or SWCNT composites as coatings for their superior conductive properties. However, the incorperation of SWCNTs into hydrogels or other polymer-based mediums can lead to nanotube aggregation and subsequent toxicity or functional inconsistency. Homogenous dispersion of SWCNTs produced by introducing water soluble polymers can help to alleviate this issue and produce a more effacious coating for interface stimulation and recording.

#### 4.3.2. Scaffolding and Coating with JBNt

The strength and elasticity, as well as other mechanical properties, of JBNt have not been as extensively studied as CNT. However, they hold a lot of potential to be incorporated into scaffolds and surface coatings to either sustain effective drug/gene delivery or promote cellular interaction [2]. 

As previously mentioned, JBNt is biocompatible and hydrophilic, with a positive zeta potential or ionic charge [2]. Not only does a positive charge offer penetration of the cellular membrane, but it can also aid in the attraction of cells to the material surface if used as a coating. Zhang et al. combined RNTs’ collagen-like structure with nanocrystalline hydroxyapatite (HA) to enhance osteoblast adhesion to titanium implants, successfully creating a biomimetic surface coating using a composite of JBNt [116]. This has potential for future applications in bone regeneration and tissue regeneration in general.

A biomimetic scaffold, or janus-base nano-matrix (JBNm), using JBNt and fibronectin (FN), was fabricated by Zhou et al. to promote human mesenchymal stem cell (hMSC) adhesion and anchorage [23]. The JBNm self-assembles through the ionic interaction between the positive charge of the JBNt and the negative charge of FN, forming a scaffold that mimics the structure of ECM and can be injected into target sites for the purpose of tissue regeneration. JBNt have also been utilized to form a layer-by-layer JBNm, hierarchically assembled via bioaffinity using JBNt, matrilin-3 [117,118], and transforming growth factor β-1 (TGF-β1). Human mesenchymal stem cells (hMSCs) were shown to prefer to anchor along the JBNm fibers, and the JBNm successfully achieved localized drug delivery through injection [119]. 

## 5. Conclusions

There is an obvious disparity between the extent of the research regarding CNTs and the research regarding JBNt, with a more large-scale and comprehensive investigation on the commercially used CNTs. While composition and synthesis methods vary drastically, CNT and JBNt both exhibit similar tubular shapes and applications in terms of drug delivery and scaffolding. The two nanotube types can be utilized for both active and passive drug and gene delivery or promote enhanced biological responses when used as a scaffold or surface coating for implants. The attraction of CNTs is the extraordinary mechanical and conductive properties, but disadvantages include the inability to synthesize completely uniform CNTs in terms of morphology and chirality, as well as the lack of biodegradability and elevated cytotoxicity with increasing amounts of the nanotubes used in vivo. JBNts, or RNTs, offer a similar approach, but use an easily synthesized DNA-mimicking structure that is highly biocompatible and bioactive, offering more biomimetic applications. JBNts also allow for not only the encapsulation of therapeutic agents, but can also be synthesized into nanopieces, which offer an effective and larger delivery vehicle for siRNA that are effective for penetrating through hard-to-reach ECM. While further research is necessary for both CNTs and JBNts to understand the full extent of their possible applications and drawbacks, they both hold great potential for future advancements in biomedical engineering and the use of nanoparticles.

## Figures and Tables

**Figure 1 ijms-23-02640-f001:**
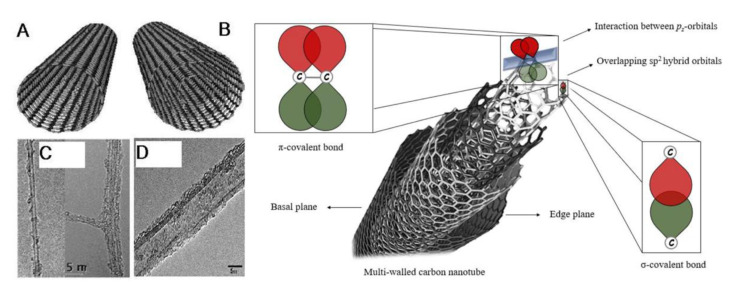
Structural schematic of (**A**) SWCNTs and (**B**) MWCNTs as well as their respective TEMs (**C**,**D**) (reproduced from [11,12,13]). MWCNT overlapping sp^2^ hybrid orbitals including sigma and pi covalent bond interactions (reproduced from [14]).

**Figure 2 ijms-23-02640-f002:**
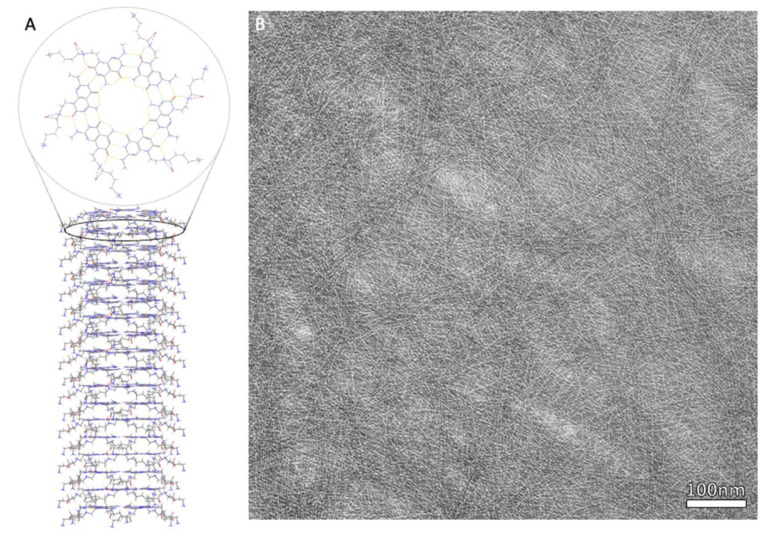
(**A**) Chemical structure of assembled JBNt comprised of hydrogen bonded rosettes, stacked via π-π bonds to form elongated tubes. (**B**) Complimentary TEM of dense JBNt network.

**Figure 3 ijms-23-02640-f003:**
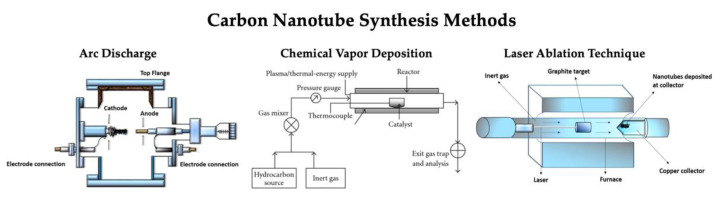
CNT synthesis methods including arc discharge method, chemical vapor deposition method, and laser ablation technique (reproduced from [31,32] to form single wall carbon nanotubes (SWCNTs) and multiwalled carbon nanotubes (MWCNTs).

**Figure 4 ijms-23-02640-f004:**
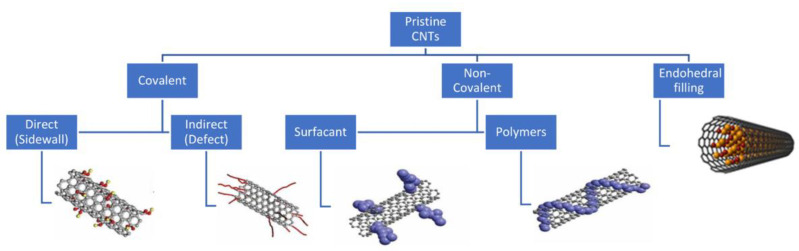
Functionalization strategies for CNTs, including covalent, noncovalent, and endohedral filling methods (reproduced from [53]).

**Figure 5 ijms-23-02640-f005:**
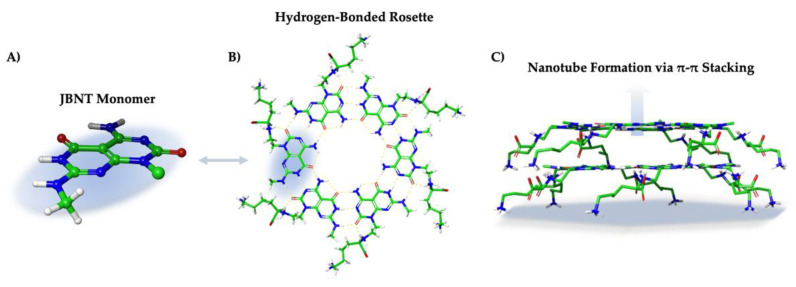
(**A**) JBNT monomer isolated from a (**B**) hydrogen-bonded rosette with amino acid side chains. (**C**) π electrons from each aromatic ring system interact with opposing rosette π electrons to self-assemble into elongated biomimetic nanotubes.

**Figure 6 ijms-23-02640-f006:**
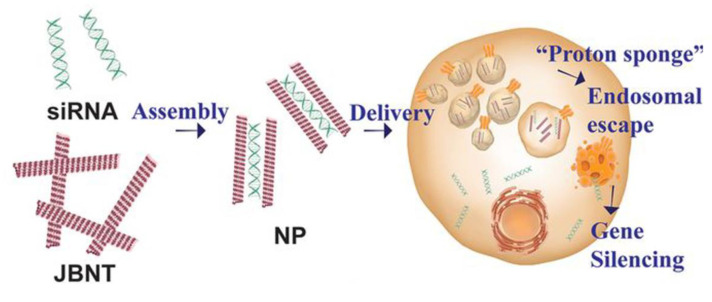
siRNA encapsulation by JBNt to form JBNp. JBNp enters the cell by micropinocytosis and undergoes endosomal escape through the “proton sponge effect” (reproduced from [93]).

## Data Availability

Not applicable.

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
