# Peer review of "Comparison between Janus-Base Nanotubes and Carbon Nanotubes: A Review on Synthesis, Physicochemical Properties, and Applications"

_ijms, 2022, doi:10.3390/ijms23052640_

Round 1

Reviewer 1 Report

Well-written and well-described paper. I didn't detect any major flaws, so my recommendation is to publish with minor reviews. 

My suggestions would be to put at least some schemes about the formation of the nanotubes, for the SWCNT, MWCNT, and the Janus-based system. Also, some more visuals can be put for the rosette assembly, and for the delivery systems.

Author Response

Thank you.

Reviewer 2 Report

This work by Griger and colleagues is a comprehensive review focused on the comparison between CNTs and JBNTs. 

Although the manuscript is very well written, the authors must incorporate several figures that illustrate significant examples.

There are three groups focused on the research of carbon nanoforms that have worked a lot with carbon nanotubes and the authors barely mention them (or do not mention them at all): the research groups of Professors Tomas Torres (https://orcid.org/0000-0001-9335-6935), Nazario Martin (https://orcid.org/0000-0002-5355-1477) and YuHuang Wang (https://orcid.org/0000-0002-5664-1849). Some selected references about these groups should incorporated. 

Furthermore, there is an interesting application of the CNTs in the area of molecular machines (see the works of Professor Emilio Perez: https://orcid.org/0000-0002-8739-2777). The use of mechanically interlocked CNTs could led to the preparation of different stimuli-responsive materials with potential applications in drug delivery (such other interlocked related materials: https://doi.org/10.1021/ja904982j). The authors should incorporate this application to the review and present their point of view on the possibilities that JBNTs will offer in this area. 

Although the authors write about some future perspectives of the application of JBNTs, I miss a more detailed opinion, not only indicated their possible use, but also how they believe it could be performed.

Strengths: 

  1. Well-written.
  2. Topic of interest.

Weak points:

  1. There are almost no figures (which makes it difficult to visualize what is explained).
  2. Missing references from some relevant authors.
  3. Some hotter applications are not addressed (I have given the example of interlocked CNTs that the authors should incorporate in the main text, but there are also many other things that could be cover in depth, such as studies of the interactions of CNTs with neurons).
  4. The authors should give an opinion based on their expertise on how they think the future applications will be reached (what should be addressed and how should it be done).

Author Response

Thank you.

Round 2

Reviewer 2 Report

I consider the revised version appropiate for publication in Molecules. However, there is a major concern. The authors incorporate figures and indicate "reproduced from reference x". They do not indicate the permissions of the corresponding editorial properly. Did the authors ask for permission? If not, this paper cannot be published until they ask for permission to the publishers and indicate this statement in the figure captions.

There is a mistake on line 414: "diffiult" should be changes by "difficult".

Author Response

Thank you.
